# Ageing-Related Neurodegeneration and Cognitive Decline

**DOI:** 10.3390/ijms25074065

**Published:** 2024-04-05

**Authors:** Irina Alafuzoff, Sylwia Libard

**Affiliations:** 1Department of Pathology, Uppsala University Hospital, 751 85 Uppsala, Sweden; sylwia.libard@igp.uu.se; 2Department of Immunology, Genetics and Pathology, Uppsala University, 751 05 Uppsala, Sweden

**Keywords:** ageing, hyperphosphorylated-τ, amyloid β-protein, α-synuclein, transactive DNA-binding protein 43, PART, ADNC, LATE, LBD/PD, ARTAG

## Abstract

Neuropathological assessment was conducted on 1630 subjects, representing 5% of all the deceased that had been sent to the morgue of Uppsala University Hospital during a 15-year-long period. Among the 1630 subjects, 1610 were ≥41 years of age (range 41 to 102 years). Overall, hyperphosphorylated (HP) τ was observed in the brains of 98% of the 1610 subjects, and amyloid β-protein (Aβ) in the brains of 64%. The most common alteration observed was Alzheimer disease neuropathologic change (ADNC) (56%), followed by primary age-related tauopathy (PART) in 26% of the subjects. In 16% of the subjects, HPτ was limited to the locus coeruleus. In 14 subjects (<1%), no altered proteins were observed. In 3 subjects, only Aβ was observed, and in 17, HPτ was observed in a distribution other than that seen in ADNC/PART. The transactive DNA-binding protein 43 (TDP43) associated with limbic-predominant age-related TDP encephalopathy (LATE) was observed in 565 (35%) subjects and α-synuclein (αS) pathology, i.e., Lewy body disease (LBD) or multi system atrophy (MSA) was observed in the brains of 21% of the subjects. A total of 39% of subjects with ADNC, 59% of subjects with PART, and 81% of subjects with HPτ limited to the locus coeruleus lacked concomitant pathologies, i.e., LATE-NC or LBD-NC. Of the 293 (18% of the 1610 subjects) subjects with dementia, 81% exhibited a high or intermediate level of ADNC. In 84% of all individuals with dementia, various degrees of concomitant alterations were observed; i.e., MIXED-NC was a common cause of dementia. A high or intermediate level of PART was observed in 10 subjects with dementia (3%), i.e., tangle-predominant dementia. No subjects exhibited only vascular NC (VNC), but in 17 subjects, severe VNC might have contributed to cognitive decline. Age-related tau astrogliopathy (ARTAG) was observed in 37% of the 1610 subjects and in 53% of those with dementia.

## 1. Introduction

In 2011, Heiko Braak and colleagues summarized the stages of neuropathologic changes (NC) seen in Alzheimer disease (AD) across the age range of 1 to 100 years [1]. In the publication, the Braak stage, based on the regional distribution of hyperphosphorylated τ (HPτ), ranging from 0 to stage VI, and the Thal phase, based on the regional distribution of the amyloid β-protein (Aβ), ranging from 0 to 5, were summarized in a diagram. This assessment was conducted on more than 2000 subjects [1,2,3,4]. The publication highlighted the existence of subjects with what has since 2014 been known as primary age-related tauopathy (PART), i.e., subjects having HPτ with an AD-related distribution but lacking concomitant Aβ pathology [5]. Prior to the recognition of PART, in 2012, two United States organizations, the National Institute of Aging and the Alzheimer’s Association, introduced new guidelines for assessing pathology in AD [6]. The recommended strategy was to assign the level of ADNC based on the Braak stage of HPτ and the Thal phase of Aβ. In addition, the neuritic plaque score defined by the Consortium to Establish Registry for AD was to be assigned [1,4,7]. The outcome yielded a range of ADNC levels, from nothing to high. In the same year, Thomas Montine and colleagues published an update to these National Institute of Aging and Alzheimer’s Association guidelines, emphasizing that besides ADNC, concomitant alterations, such as Lewy body disease (LBD)-NC, an α-synuclein (αS) pathology, and vascular (V) NC, need to be assessed and registered [8]. The assessment of the severity of LBD-NC was based, as above, on the regional distribution of αS pathology, which ranges from brainstem involvement to neocortical [9,10]. The inclusion of concomitant pathologies when assessing ADNC was important, as it was recognized that MIXED-NC is a common observation in the brains of the aged [11]. Thereafter, in 2014, Keith Josephs and colleagues reported that the transactive DNA-binding protein 43 (TDP43) played a critical role in the setting of ADNC, in addition to LBD-NC and VNC [12]. In the same year, a staging strategy for TDP43 pathology, also based on the regional distribution of the alteration, was introduced, and it was updated in 2016 [13,14]. To further emphasize the significance of TDP43 pathology in the aged, Peter Nelson and colleagues introduced a new neuropathological entity, i.e., limbic-predominant age-related TDP43 encephalopathy (LATE), which was updated in 2022 [15]. Furthermore, Gabor Kovacs and colleagues defined and published a standardized evaluation strategy for ageing-related tau astrogliopathy (ARTAG) [16]. In 2020, we reported that MIXED-NC is the most common cause of cognitive decline (CD) in the aged [17]. Following this, in 2021, McAleese and colleagues reported that concomitant neurodegenerative pathologies contribute to the progression from mild cognitive impairment (MCI) to dementia [18]. Regarding VNC, the Vascular Cognitive Impairment Neuropathological Guidelines suggested a method for evaluating the impact of VNC on CD, incorporating the assessment of infarcts, arteriosclerosis, and cerebral amyloid angiopathy (CAA) [19].

In summary, since 2011, a substantial number of publications dealing with what to assess and how in the brains of the aged have been published. Moreover, in addition to ADNC, conditions such as PART, ARTAG, and LATE-NC have been defined.

The primary objective of this study was to examine all pathologies listed above in a standardized manner by applying reproducible automatic staining techniques. The study was conducted on an unselected sample of aged individuals, including clinically unimpaired subjects and those diagnosed with dementia. The aim was to evaluate the neurodegenerative process in the ageing brain, to determine the incidence of conditions such as ADNC, PART, LATE-NC, LBD-NC, and ARTAG, and to examine the incidence of MIXED-NC in a large sample of deceased individuals with ages at death ranging from 41 to 102 years.

## 2. Results

During the 15-year-long study period, 36,034 deceased individuals were admitted to the morgue at Uppsala University Hospital, and clinical autopsies were performed on 4389 (12%) cases. A neuropathological assessment was carried out on 1630 (37%) of the 4389 subjects. Thus, the brains of 5% of all 36,034 deceased individuals that were admitted to the morgue were assessed (Flowchart in Table 1).

One of the selection criteria for this study was that the age at death should be ≥41 years. The second criterion was the observation of HPτ pathology in the brain tissue, with a distribution as described by Braak and colleagues [1,3].

Out of the available 1630 cases, 1610 fulfilled the first criterion of being ≥41 years at death. The mean age ± SE was 76.2 ± 0.3 years, ranging from 41 to 102 years at death. Of these cases, 58% were males and 42% females, and 293 (18%) subjects displayed dementia based on clinical records. Among the 1610 subjects, HPτ was not observed in the locus coeruleus (LC) or within the distribution as described for PART or ADNC in 34 (2%) subjects, comprising 17 males or females. The mean age ± SE was 68.9 ± 1.8 years, with a range of 41–87 years, i.e., significantly (MWU = 0.000) younger compared to the rest of the sample. A total of 5 (15%) of the 34 subjects had been diagnosed with dementia based on clinical records. In 14 of these subjects (7 males and 7 females), no brain alterations were seen. In 3 out of these 34 subjects, Aβ was observed either only in the neuropil (female, 58 years at death), only as CAA (female, 67 years at death), or both (male, 72 years at death). None of these 17 subjects were reported as having dementia. The remaining 17 subjects, 9 males and 8 females, exhibited disease-related NC, as summarized in Table 2 (no dementia) and Table 3 (with dementia) under the column “no HPτ”. Alterations in line with primary tauopathy, i.e., progressive supranuclear palsy/corticobasal degeneration (PSP/CBD), were the most common alterations seen in both subjects without and subjects with dementia (32%). In 9 (27%) of these 34 subjects, ARTAG was observed. These 34 subjects have been excluded from the following analysis (flowchart in Table 1).

In the remaining 1576 subjects (98% of all ≥41 years at time of death), HPτ pathology was observed within the distribution as described by Braak and colleagues in 2011 [1]. The mean age ± SE was 76.4 ± 0.2 years, ranging from 48 to 102 years at death. Of the 1576 subjects, 911 (58%) were male and 665 were female. Dementia was registered in 288 (18%) subjects.

In 155 (10%) of these 1576 subjects, significant NC and HPτ pathology limited to LC (*n* = 14) or in line with ADNC/PART (*n* = 82/59) in Braak stages I–II were observed (Table 1). The mean age ± SE was 78.7 ± 0.7 years, comprising 101 (65%) males and 54 females. Forty (26%) subjects were reported as having dementia. Twenty-two subjects (55%) were males and eighteen were females; mean age ± SE was 76.9 ± 1.6 years. In 113 (73%) out of these 155 subjects, LBD-NC pathology with αS Braak stage ≥ 4 was observed. In 26 (17%) subjects, LATE-NC pathology with Josephs TDP43 stage ≥ 3 was observed. Additionally, 21 subjects displayed NC as seen in FTLD with TDP, HPτ, or FUS [20]. Dementia was primarily attributed to LBD-NC or FTLD. Detailed NC are seen in Table 2 and Table 3, under the columns + HPτ/LC, +PART/L, and +ADNC/L. In 62 (40%) of the 155 subjects, ARTAG was observed. These 155 subjects have been excluded from the following analysis (flowchart in Table 1).

In 237 (17%) of the remaining 1421 subjects, HPτ pathology was limited to the LC. Among these, 158 (67%) were males and 79 were females, with a mean age ± SE of 67.6 ± 0.6 and an age range of 49 to 94 years. This group was significantly (MWU = 0.000) younger compared to subjects in Braak stages I–VI. None of these subjects displayed CD. In 60 (25%) cases, concomitant Aβ pathology was observed, with 40 cases showing Aβ only in the neuropil, four cases showing it only as CAA, and 16 cases showing Aβ in both the neuropil and as CAA. Concomitant with the HPτ pathology limited to the LC, LBD-NC with αS in Braak stages 1–3 was observed in 10 (5%), LATE-NC with TDP43 in Josephs stage 1–2 was observed in 22 (10%), and both LBD and LATE-NC were observed in one subject. ARTAG was noted in 32 (14%) of these subjects. The NC observed in these subjects are summarized in Table 2 under the column HPτ/LC. These 237 subjects have been excluded from the following analysis.

In summary, HPτ pathology was observed in 1184 subjects, ranging from Braak stages I–VI, comprising 652 (55%) males and 532 females. The average age at death ± SE was 77.8 ± 0.3 years, ranging from 48 to 102, and 248 subjects (21%) had suffered from dementia during their lifetime. Out of the 1184 subjects, 822 (69%) displayed ADNC and 362 (31%) displayed PART. In Table 4, demographics and the incidence of main NC observed are summarized.

Subjects with ADNC were significantly older compared to those with PART, and dementia was significantly more common in the ADNC group. It is noteworthy that subjects with HPτ limited to LC were significantly (MWU = 0.000) younger compared to subjects with PART. Alterations such as ARTAG, LATE-NC, LBD-NC, CAA, and VNC were significantly more common and severe in subjects with ADNC compared to those with PART. In contrast, argyrophilic grains (Ag) were significantly more common in subjects with PART compared to those with ADNC (Table 4).

### 2.1. Subjects in the Fifth Decade

There was one male subject without dementia who was 48-years-old at the time of death. His brain showed a low level of HPτ pathology without any other protein alterations, fulfilling the definite criteria for PART.

In Table 5, demographics and the incidence of the main NC (Figure 1) observed are summarized for subjects in their 6th, 7th, 8th, 9th, and 10th decades. There were seven subjects in the 11th decade. In Table 5, and for statistical purposes, they are included in the group of the 10th decade. Detailed NC for all subjects is listed in Table 2 and Table 3.

### 2.2. Subjects in the Sixth Decade

There were 36 subjects in this group, comprising 21 males and 15 females. Among these, 56% displayed ADNC, and two (6%) of the 36 subjects, both with ADNC, had dementia. One male subject, 50-years-old at the time of death, had AD with familial clustering, and one subject—age at death, 59 years—had Down’s syndrome. Both subjects displayed a high level of ADNC, whereas concomitant alterations were sparse. Only the male subject dying at the age of 50 years displayed ADNC with concomitant mild LATE-NC. In the 34 cognitively unimpaired subjects, the incidence of concomitant protein alterations was also low. Most of the subjects (85%) displayed pure ADNC/PART (14/15, respectively), while in five (15%) subjects, as shown in Table 5 and Figure 1, concomitant lesions were observed (three ADNC/LATE-NC; one PART/LATE-NC; one ADNC/LBD-NC). One subject with ADNC and severe VNC did not have dementia. ARTAG was observed in 3 (8%) of these 36 subjects (ADNC/PART = 2/1).

### 2.3. Subjects in the Seventh Decade

There were 179 subjects in this group, comprising 112 males and 67 females. Out of these, 60% displayed ADNC. In the seventh decade, when comparing ADNC with PART, the former cases displayed concomitant alterations, such as LATE-NC and CAA, significantly more frequently (Table 5, Figure 1). Dementia was diagnosed in nine (5%) subjects, all of whom displayed ADNC pathology. In three subjects (33%), the CD was attributed to pure ADNC. Three (33%) subjects displayed concomitant LATE-NC, while three displayed both concomitant LATE- and LBD-NC. Out of the 170 cognitively unimpaired subjects, 116 (68%) displayed pure ADNC/PART (62/54, respectively). In two subjects with PART, concomitant CAA was observed; 30 (18%) subjects displayed concomitant LATE-NC (ADNC/PART = 22/8); 14 (8%) concomitant LBD-NC (ADNC/PART = 10/4); and in 6 (4%) subjects, both LATE-NC and LBD-NC were observed (ADNC/PART = 4/2). Additionally, one subject showed NC in line with ADNC/PSP/LATE-NC, and another subject showed NC in line with PART/Ag/LATE-NC. There was one subject with dementia and one subject without dementia with ADNC/LATE-NC in Josephs stage ≤ 3 and concomitant severe VNC. These two subjects differed in that the subject with dementia displayed intermediate ADNC, while the subject without dementia showed a low level of ADNC. ARTAG was observed in 45 (25%) of these subjects (ADNC/PART = 27/18)

### 2.4. Subjects in the Eighth Decade

There were 447 subjects, comprising 242 males and 215 females, in this group, and 65% of them displayed ADNC. In the eighth decade, when comparing ADNC with PART, the former cases displayed concomitant alterations, such as LATE-NC, LBD-NC, CAA, and VNC, significantly more frequently as well as more severely. Dementia was diagnosed in 77 (17%) out of 447 subjects, significantly more often compared to those in their seventh decade. In 13 (17%) of the subject with dementia, CD was attributed to pure ADNC. Thirty (40%) of the subjects with dementia displayed concomitant LATE-NC (ADNC/PART = 28/2). Seven (9%) subjects displayed concomitant LBD-NC (ADNC/PART = 6/1), and in 24 (31%) subjects, both concomitant LATE-NC and LBD-NC were observed (ADNC/PART = 21/3). One subject displayed NC in line with ADNC/FTLD, one in line with ADNC/MSA, and one in line with PART/Ag. Severe VNC was observed in 2 out of 77 (3%) subjects, which might have eventually contributed to CD. Out of the 370 cognitively unimpaired individuals, 228 (61%) displayed pure ADNC/PART (129/99). In 9 PART cases, concomitant CAA was observed, while in 87 (24%) subjects, concomitant LATE-NC (ADNC/PART = 57/30) was present. In 21 (5%) subjects, concomitant LBD-NC (ADNC/PART = 16/5) was observed, and in 21 (6%) subjects, both LATE-NC and LBD-NC were observed (ADNC/PART = 17/4). There were two subjects without dementia with NC in line with ADNC/FTLD, one in line with ADNC/PSP/LATE-NC, and one in line with PART/Ag. In 15 (4%) of the 370 cognitively unimpaired subjects, severe VNC was observed. ARTAG was observed in 165 (37%) of these subjects (ADNC/PART = 113/52)

### 2.5. Subjects in the Ninth Decade

There were 400 subjects, comprising 214 males and 186 females, in this group, and 77% of them displayed ADNC. In the ninth decade, when comparing ADNC and PART, the former cases displayed concomitant alterations, such as LATE-NC, LBD-NC, CAA, and VNC, significantly more frequently as well as more severely, in line with the subjects in the eighth decade. Dementia was diagnosed in 126 (32%) subjects, significantly more often when compared with those in the eighth decade. In 25 subjects (20%), the CD was attributed to ADNC only. In 55 (44%) subjects with ADNC, concomitant LATE-NC was observed. Additionally, 15 (10%) subjects displayed concomitant LBD-NC (ADNC/PART = 14/1), and in 26 (22%) subjects, ADNC was observed with both concomitant LATE-NC and LBD-NC. There were four subjects with NC consistent with ADNC/FTLD, and one subject with ADNC/PSP/LATE-NC. In 11 (9%) out of the 126 subjects with dementia, there was evidence of severe VNC, which might have contributed to the CD. In 130 (47%) out of the 274 cognitively unimpaired subjects, pure ADNC/PART (80/50) was observed, and in 7 PART cases, concomitant CAA was observed. In 80 (30%) subjects, concomitant LATE-NC (ADNC/PART = 59/21) was noted, while in 24 (9%) subjects, concomitant LBD-NC (ADNC/PART = 16/8) was observed. Additionally, in 25 (9%) subjects, both LATE-NC and LBD-NC were observed (ADNC/PART = 21/4). Furthermore, there was one subject with NC in line with ADNC/FTLD, four subjects with concomitant Ag (ADNC/PART = 1/3), and in four subjects, concomitant PSP/CBD-NC (ADNC/PART = 3/1) was seen. In 15 (6%) of these 274 cognitively unimpaired subjects, severe VNC was noted. ARTAG was detected in 190 (48%) of these subjects (ADNC/PART = 147/43).

### 2.6. Subjects in the 10–11th Decade

There were 121 subjects, comprising 62 males and 59 females, who were aged ≥ 90 years at the time of death, and 79% of them displayed ADNC. In this group, significant differences in the incidence of concomitant alterations were observed only for severe LATE-NC and CAA when comparing ADNC and PART. Dementia was registered in 34 (28%) subjects. Only four subjects (12%) showed pure ADNC as the cause of the CD. Eleven (32%) subjects displayed concomitant LATE-NC (ADNC/PART = 10/1), three (9%) subjects displayed concomitant LBD-NC (ADNC/PART = 2/1), and fifteen (44%) subjects with ADNC displayed concomitant LATE-NC and LBD-NC. In one subject with dementia, NC in line with PART/Ag/LATE-NC was seen. In 2 (6%) out of 34 subjects, severe VNC contributed to the CD. In 27 (31%) out of the 87 cognitively unimpaired subjects, pure ADNC/PART (20/7) was observed, and concomitant CAA was noted in one PART subject. In 45 (52%) subjects, concomitant LATE-NC (ADNC/PART = 35/10) was observed, and in 4 (5%) subjects, concomitant LBD-NC (ADNC/PART = 3/1, respectively) was seen. Additionally, in nine (10%) subjects, both concomitant LATE-NC and LBD-NC (ADNC/PART = 6/3) were detected. In one subject, NC in line with ADNC/Ag was observed. In 7 (8%) out of the 87 cognitively unimpaired subjects, severe VNC was seen. ARTAG was observed in 84 (70%) of these subjects (ADNC/PART = 68/16). Included in the sample of 121 subjects in 10th decade, there were seven subjects in their 11th decade, comprising two males and five females. None had displayed dementia during their lifetime. Five subjects displayed intermediate levels and two displayed a low level of ADNC/PART. Four displayed ADNC/LATE-NC, two displayed ADNC/LATE-NC/LBD-NC, and one displayed ADNC/Ag.

### 2.7. Clinical Signs of Dementia

Out of the 1610 available cases of individuals aged ≥41 years at the time of death, 293 subjects were found to have dementia. The mean age at death was ±SE, 80.8 ± 0.5 years, and 54% of the subjects were females (Table 3). Based on the medical records, none of the subjects had been a member of a pharmacological intervention directed to Aβ. The incidence of dementia increased significantly (PCS <0.001) with each decade (5th decade (0%), 6th decade (5%), 7th decade (5%), 8th decade (16%), 9th decade (29%), and 10th decade (27%)), particularly starting from the 8th decade. From the 7th decade, CD was more common in females (male/females, 7^th^ decade (2/10%), 8th decade (11/22%), 9th decade (26/33%), and 10th decade (23/31%)). Five subjects lacked HPτ pathology; in four subjects, HPτ pathology was restricted to LC, and 37 subjects displayed sparse ADNC/PART (Braak stages I–II). In the remaining 247 (84%) subjects out of 293, the ADNC/PART pathology was assessed to be severe enough, i.e., intermediate or high level, for the clinical symptoms of dementia. In Table 3, the NC are summarized for all subject with dementia. The most common cause of dementia was assessed to be ADNC, with concomitant LATE-NC observed in 33% of the subjects. This group was significantly older compared to subjects displaying pure ADNC (MW *p* > 0.022). There were three subjects with PART/LATE-NC, one fulfilling the definite diagnostic criteria for tangle-predominant dementia and two for LATE. The number of subjects with dementia with ADNC increased (PCS = ns) with age (6th decade (40%), 7th decade (53%), 8th decade (62%), 9th decade (62%), and 10th decade (80%)), while the number of subjects with pure ADNC, accounting for 16% of all cases, decreased with age (PCS = ns) (6th decade (20%), 7th decade (20%), 8th decade (15%), 9th decade (17%), and 10th decade (11%)). In line with ADNC/LATE-NC, ADNC with both concomitant LATE-LC and LBD-NC was relatively common (24%). Eight subjects displayed PART/LATE-NC/LBD-NC. In 17 subjects (6%), severe VNC might have contributed to the CD. In these 17 cases, the significant NC were ADNC in two subjects, ADNC/LATE-NC in six, ADNC/LBD-NC in two, ADNC/LBD-NC/LATE-NC in six, and in one, HPτ in LC/FTLD-NC.

### 2.8. Concomitant Pathologies

Out of the 1184 subjects with ADNC or PART (Table 2, Table 3 and Table 4, Figure 2), 79% of the individuals with dementia displayed concomitant pathologies compared to 40% of the significantly younger cognitively unimpaired subjects. The most common concomitant alteration was LATE-NC, observed in 30% of the 1184 subjects, followed by LATE-NC/LBD-NC in 11% and LBD-NC in 8% of the subjects. The percentage of subjects with dementia and concomitant alterations (Figure 1) increased with age: (6th decade (50%), 7th decade (67%), 8th decade (82%), 9th decade (80%), and 10th decade (88%)). A significant increase (PCS < 0.001) in the number of subjects with concomitant alterations was observed with increasing age for the unimpaired group (6th decade (15%), 7th decade (30%), 8th decade (36%), 9th decade (49%), and 10th decade (65%)). A significant correlation was observed between age and the extent of assessed tissue alterations for HPτ, Aβ, TDP43, and αS in the cohort of 1184 subjects (Table 6). Similarly, a strong correlation was observed between the levels of HPτ, Aβ, TDP43, and αS, particularly between HPτ and Aβ or HPτ and TDP43. When the sample was divided into PART and ADNC groups, the outcome did not change significantly. When the sample was divided into individuals with dementia and those without, the outcome changed significantly, especially for those with dementia and particularly in terms of the correlation between age and the degree of protein alterations (Table 6). A weak association was observed between the severity of HPτ (low, intermediate, high levels of ADNC/PART) and severe VNC (PCS 0.034). A significant association was observed between severe VNC and Aβ and TDP43 (FE < 0.001 and 0.002, respectively). In Figure 1, in the Venn diagram, the relationship between the altered proteins is visualized for the whole sample (Table 2 and Table 3), as well as for the subjects with ADNC or PART (Table 4).

#### Limbic-Predominant Age-Related TDP Encephalopathy (LATE)

LATE-NC was observed in the brains of 35% of the 1610 subjects aged ≥41 at the time of death. When comparing subjects with ADNC and PART (Table 4), LATE-NC was observed more frequently (FE < 0.001) in ADNC (47%) than in PART (26%) and more frequently (FE < 0.001) in the brains of individuals with dementia (69%) compared to individuals without dementia (34%). The prevalence of LATE-NC in the brains of subjects increased significantly (PCS < 0.001) with age (6th decade (10%), 7th decade (19%), 8th decade (31%), 9th decade (42%), and 10th decade (65%)) and with the level of ADNC (low 27%, intermediate 62%, high 79%). LATE-NC was never observed without concomitant HPτ; i.e., in four subjects, HPτ was seen as Ag or PSP/CBD. Concomitant LATE-NC was observed in 27 subjects with HPτ limited to LC, whereas most subjects with LATE-NC displayed ADNC or PART. A strong positive correlation (Table 6) was observed between the severity of TDP43 and HPτ.

### 2.9. Age-Related Tau Astrogliopathy (ARTAG)

ARTAG was observed in the brains of 37% of the 1610 subjects aged ≥41 at the time of death, in 53% of subjects with dementia, and in 33% of subjects without signs of dementia. Out of the 1184 subjects with ADNC or PART (Table 4), 41% displayed ARTAG. The number of subjects with ARTAG increased significantly (PCS < 0.001) with age (6th decade (8%), 7th decade (25%), 8th decade (37%), 9th decade (48%), and 10th decade (69%)). A significantly (FE < 0.021) higher number of subjects with ARTAG was observed in cases with ADNC (43%) compared to subjects with PART (36%). A significant (PCS < 0.01) increase in the number of subjects with ARTAG was also observed with the rise in the level of ADNC (low 32%, intermediate 53%, high 50%). In accordance with the above, ARTAG was observed more frequently in subjects with CAA (50%) compared to subjects without CAA (37%), in subjects with VNC (45%) compared to those without VNC (39%), and in subjects with severe VNC (61%) compared to those without severe VNC (40%) (FE < 0.001; <0.001; =0.005, respectively).

### 2.10. Cerebral Amyloid Angiopathy (CAA)

CAA was observed in the brains of 28% of the 1610 subjects aged ≥41 at the time of death, in 51% of subjects with dementia, and in 23% of subjects without signs of dementia. When comparing subjects with ADNC and PART (Table 4), CAA was observed more frequently (FE < 0.001) in ADNC (43%) compared to PART (10%), and it was also more common (FE < 0.001) in the brains of individuals with dementia (55%) compared to individuals without dementia (27%). The observation of CAA in the brains of subjects increased significantly (PCS < 0.001) with age (6th decade (14%), 7th decade (21%), 8th decade (30%), 9th decade (40%,) and 10th decade (44%)) and with the level of ADNC (low 33%, intermediate 44%, high 72%).

### 2.11. Vascular Neuropathologic Change (VNC)

VNC was observed in the brains of 62% of the 1610 subjects aged ≥41 at the time of death, in 71% of subjects with dementia, and in 59% of subjects without signs of dementia. Severe VNC was observed in 62 (4%) of the 1610 subjects aged ≥41 at the time of death and in 17 (6%) of the 293 subjects with dementia. When comparing subjects with ADNC and PART (Table 4), VNC was observed (FE < 0.001) in subjects with ADNC (69%) significantly more frequently compared to PART (52%) in line with the severe VNC, which was more common (FE < 0.001) in subjects with ADNC (6%). VNC increased significantly (PCS < 0.001) with age (5th decade (33%), 6th decade (50%), 7th decade (47%), 8th decade (59%), 9th decade (71%), and 10th decade (77%)), as well as severe VNC (PCS <0.001) (6th decade (1%), 7th decade (1%), 8th decade (3%), 9th decade (6%), and 10th decade (7%)). Alterations in line with etat crible/etat lacunaire in striatum were seen in 24% of the 1610 subjects aged ≥40 at the time of death and increased significantly (PCS < 0.001) with age (5th decade (17%), 6th decade (12%), 7th decade (19%), 8th decade (21%), 9th decade (32%), and 10th decade (29%)). Gender or symptoms of dementia did not influence the incidence of VNC.

## 3. Discussion

In this paper, we had the opportunity to assess NC in 1610 brains obtained from subjects aged ≥41-years-old at the time of death (Table 1). This cohort represents approximately 5% of all deceased individuals admitted to the local morgue during a 15-year period, with an autopsy frequency of 12%. The neuropathological assessment was conducted in about 37% of all autopsy cases.

### 3.1. Autopsy Service and following Neuropathological Assessment

It has been reported that a sparse number of neuropathological assessments are related to the low frequency of autopsies in the Western world, estimated, overall, to be only a few percent [21]. In Sweden, and at our University hospital, autopsies are carried out following a referral from a caregiver, with the primary objective being the assignment of a cause of death. The postmortem neuropathological assessment is carried out when subjects have displayed neurological deficits, there were queries regarding brain function, or the subjects are old enough to be expected to display age-related NC that might have influenced the mode of death [1]. Recently, it was published that autopsies are performed nationwide in Sweden on 12.6% of all deceased individuals, in line with our percentage, which is high when compared to the estimated frequency worldwide [21,22]. Based on observations in the nationwide study, autopsies were preferably carried out on male and younger subjects, whereas being female or dying with dementia seldom led to the performance of an autopsy [22]. Consistent with the reported nationwide outcome, even locally, we noted that more males (58%) compared to females were assessed postmortem. Notably, only 37% of our autopsy cases were in the 9th-to-11th decades, an observation in line with what was reported for the whole nation; i.e., young subjects are preferably assessed postmortem [22]. When studying ageing-related alterations, this is a worrisome selection bias in a country with a current mean survival age for females around 85 and for men around 81 years (Statistics Sweden). It has been reported that there are substantial discrepancies between autopsy results and pre-mortal clinical diagnoses [23]. Based on the above, one needs to be cautious when interpreting outcomes from similar postmortem studies—are they representative of the whole population?

### 3.2. Clinical Signs of Cognitive Decline, Mild Cognitive Impairment, and Dementia

According to the medical records, about 18% of our subjects had displayed dementia during their lifetime, with an age at death ≥41 years and a mean ± SE of 74.1 ± 0.4 years at death. In 2019, the Swedish “Demenscentrum” estimated that there were approximately 150,000 individuals with dementia in the country (7th decade (2%), 8th decade (10%), and 9th decade (20%) of the population) with an estimated annual increase in cases ranging from 20,000 to 25,000. In our sample, the incidence of dementia increased with each decade, most dramatically from the 8th to the 9th and from the 9th to the 10th decades (6th decade (5%), 7th decade (5%), 8th decade (16%), 9th decade (29%), 10th decade (28%)). This outcome is somewhat higher compared to what was reported in 2019 by the “Demenscentrum”. This difference might be related to the selection bias discussed above, as relatively few aged individuals come to autopsy and undergo subsequent neuropathological assessment [21,22]. Another factor that probably influences both our data and those found in the Swedish National Patient Registry is whether all subjects with mild to moderate CD are indeed registered by the healthcare system, i.e., whether a clinical diagnosis of CD has been assigned and registered in the Swedish National Patient Registry by the healthcare provider. We did not separate subjects with MCI, as the data in the medical records were considered unreliable. Not all subjects in our sample had met with a geriatrician, neurologist, psychiatrist, or neuropsychologist during their lifetime. The diagnosis of CD was often given by a general practitioner, and this diagnosis was generally based on clinical presentation only. Furthermore, the observation of mild and even moderate CD might be hampered by individuals’ social situations; specifically, 35% of the population aged ≥ 60 years in Sweden live alone (Statistics Sweden), and some of our deceased subjects had sparse or hardly any contact with their primary caregivers. Thus, our non-dementia group most likely include several subjects with at least MCI, but some might even have had symptoms of a more severe CD that had been overlooked. As many as nine subjects in our “subjects without dementia” cohort displayed a high level (Braak stage V) of ADNC, and in these subjects, concomitant NC were observed. This level of ADNC, in general, suggests that clinical symptoms of CD should have been present. Furthermore, there were 75 subjects without dementia with HPτ pathology in Braak stage IV (ADNC/PART = 70/5). Even in these non-dementia subjects, concomitant alterations were common. Pathology in Braak stage IV indicates involvement of the neocortex; thus, CD symptoms would have been expected to some extent. The prodromal, preclinical state in a subject with neurodegeneration has been considered to be related to what is called cognitive resilience, which varies across individuals [24]. This outcome, however, suggests that a number of subjects in our non-dementia group might have had MCI, which may have been overlooked during their lifetime. Overall, based on the observed NC the incidence of CD might be higher than reported by the healthcare providers. The limited resources among the primary caregivers, such as a sparse level of assessments (i.e., no assessment of cerebrospinal fluid, no imaging) and the limited number of postmortem assessments of brain pathologies (i.e., few autopsies), leads to the sad conclusion that the statistical data on ageing-related neurodegeneration might be more or less unreliable.

### 3.3. Hyperphosphorylated τ and β-Amyloid

When assessing NC in a sample of 1610 subjects with an age at death of ≥41 years, it was revealed that only 14 subjects (1%) lacked any of the assessed alterations. Notably, these subjects were young, with a mean age at death ± SE of 59.1 ± 2.4 years, and an age range of 41 to 76 years. In 12 subjects, HPτ pathology was observed, in line with the NC of PSP/CBD or Ag. These findings strongly support the notion that HPτ pathology, as previously pointed out by Braak and colleagues, is a very common alteration in the aged brain [1].

There were three subjects lacking HPτ pathology but displaying Aβ in the neuropil and/or as CAA, and six subjects with HPτ limited to LC with Aβ in the neuropil and/or as CAA. Overall, Aβ was observed in <1% of the subjects without dementia lacking concomitant HPτ pathology. When compared with subjects with HPτ pathology in Braak stages I–VI, Aβ was observed in 65% of all subjects, 91% of subjects with and 59% of subjects without dementia. The incidence of detecting Aβ in the brain increases significantly with age (6th decade (46%), 7th decade (52%), 8th decade (62%), 9th decade (76%), and 10th decade (79%)) as previously described by Braak and colleagues [1]. The observation that 46% of subjects in the sixth decade already display Aβ in their brain is worrisome, as it is presumed that anti-Aβ treatment may need to be initiated before this protein alteration is observed.

### 3.4. Hyperphosphorylated τ in the Locus Coeruleus (LC)

The LC is the site of noradrenaline synthesis and plays a major role in regulating autonomic function, arousal, attention, and neuroinflammation. Alterations in the LC have been linked to AD but also seem to occur in ageing [25]. In our study, as many as 211 out of 1610 (13%) subjects exhibited HPτ pathology limited to the LC, constituting probably the largest worldwide cohort of this kind. Already in 2011, Braak and colleagues emphasized that the LC is affected by HPτ pathology in ageing, and there has been some debate about whether the LC is the site of initiation of this protein alteration or if the LC is merely secondarily affected [1,26,27]. That 13% of subjects display HPτ pathology only in the LC certainly suggests that the LC should be considered, in some cases, as a potential initiation region regarding ADNC/PART. Notably, it cannot be excluded that some of these subjects with HPτ in the LC might evolve into cases with other tauopathies such as PSP rather than ADNC/PART [28]. We did not assess whether the HPτ in LC was 3R or 4R tau or both and this certainly needs to be carried out. The role of the LC in the evolution of ADNC/PART has been debated as there are hardly any studies where the incidence of LC involvement with HPτ pathology has been systematically assessed in a large cohort including subjects with dementia and the cognitively unimpaired [29,30]. Eventual discrepant assessment outcomes are primarily related to selection bias, i.e., the selection of brains for assessment, and sampling strategy, i.e., the mode of taking specimens for analysis. In this study, neuropathological assessment was carried out on 5% of all subjects in the morgue of the University Hospital, the sample was consistently taken centrally from the pons, and the LC was always visualized, with the section stained with antibodies directed to HPτ. We assessed all 1610 subjects aged ≥41 years at the time of death and observed that subjects with HPτ solely in the LC were significantly younger compared to subjects with PART or ADNC, but also significantly younger than subjects with HPτ Braak stages I–II (MWU = 0.000). The age differences certainly suggest that HPτ can be observed only in the LC before the involvement of the entorhinal cortex if a progression of pathology is presumed [27,31]. An early involvement of the LC with HPτ is interesting from a clinical point of view, as alterations in sleep patterns or mood could eventually be registered as signs of changes in the function of the LC [32,33,34]. Consistent with the above, studies have suggested that early-life depression can be seen as a risk factor for dementia and that sleep disturbances might increase the risk of dementia [35,36]. Furthermore, it was recently reported that MRI detected deterioration of the LC in the prodromal stages of the disease [37]. In very few subjects with HPτ in LC, concomitant Aβ was detected, making these subjects, if identifiable, an eligible target group when choosing subjects for anti-Aβ treatment.

### 3.5. ADNC and PART

The remaining 1325 out of the 1610 subjects displayed HPτ pathology within Braak stages I–VI. Moreover, 32% displayed HPτ but lacked Aβ pathology (i.e., PART), and 68% displayed HPτ and Aβ (i.e., ADNC), indicating that ADNC is indeed more common compared to PART within this age frame. In 141 out of these 1325 subjects, other alterations, such as PSP/CBD, LBD-NC, FTLD, and LATE-NC, were observed, and the low level of PART/ADNC was assessed as a less significant concomitant alteration. In addition to the 141, there were 14 subjects with HPτ limited to the LC; thus, in total, 155 subjects displayed substantial concomitant alterations other than HPτ limited to the LC or ADNC/PART with HPτ in Braak stage Í–II (Table 1). Of the 155 subjects, 40 displayed CD and are listed in Table 3. The most common significant alteration was LBD-NC (75%) with αS pathology Braak stage ≥4, and FTLD-(17%). The most common cause of CD was in line with the above: LBD-NC and FTLD. Most of the subjects with LBD-NC lacking CD displayed Parkinson disease. One of the objectives of this study was to compare subjects with PART to subjects with ADNC; in order to do so, we excluded these 155 subjects from the analysis. ADNC was the most common (69%) alteration among the 1184 subjects available, and the intermediate and low level of ADNC were each represented by some 30% of all subjects. In the PART group, most subjects (*n* = 301) were in Braak stages I–II. Sixty of these subjects displayed Braak III–IV, and one subject was in Braak stage V. Only eight subjects with dementia (3%) out of 293 displayed pathologies consistent with tangle-predominant dementia (one in Braak stage V, two in Braak stage IV, five in Braak stage III), and concomitant alterations were observed in seven of these eight subjects (*n* = 3 with LATE-NC, *n* = 3 with LBD-NC, and *n* = 2 with both). As expected, tangle-predominant dementia was an unusual subtype of tauopathies [38]. There has been a debate about whether PART is indeed a separate disease entity or merely a stage before the development of ADNC [39]. Interestingly, when comparing the mean age at death between ADNC and PART in Braak stages I–II or in Braak stages III–IV, no significant differences were observed, suggesting that the significant age difference registered between all ADNC and PART cases is primarily related to the subjects in Braak stages V–VI, who are the oldest. Moreover, PART was observed in both the 9th and 10th decades, even though fewer cases were detected by each decade—4% of all PART cases in the 6th decade, 20% in the 7th decade, 43% in the 8th decade, 26% in the 9th decade, and 7% in the 10th decade. These observations suggest that some PART cases, though maybe not all, might represent a different disease when compared with ADNC. Hence, PART might not merely be a stage in the evolution of ADNC. However, it is impossible to be dogmatic on this topic, as the number of subjects with PART might indeed be further reduced with longer survival. In our series, we had seven subjects with an age at death ≥100, and only one was assigned with PART, i.e., lacked Aβ in the brain. It has been reported that only some subjects with MCI develop dementia over time [40]. This clinical observation might be related to the causative type of pathology in the brain, i.e., whether it is ADNC or PART that is the cause of MCI. If the subject displays a low-to-intermediate level of HPτ without concomitant Aβ, LATE-NC, or αS, the progression of NC might be extremely slow. In our series, it would include 226 non-subject with dementias, accounting for 17% of all non-dementia cases.

### 3.6. Concomitant Pathologies

Concomitant pathologies in AD are often neglected in clinical settings, as they are not currently assessable during life. Notably, even in neuropathological settings, not all currently known concomitant alterations are always taken into consideration, and publications prior to 2014 do not include one of the major concomitant alterations, namely, TDP43 [41]. The definite diagnosis of AD is generally given in line with the level of ADNC pathology, as recommended in 2012 (i.e., none, low, intermediate, and high levels), while information about eventual concomitant pathologies might be lacking [6,8,42]. Harmonization of a study cohort for age (as all alterations increase with age), primary NC (ADNC vs. PART) and concomitant NC (LATE-NC, LBD-NC, VNC), as shown by us here and as visualized in the Venn diagrams in Figure 1 and Figure 2, reduces the number of “similar” cases significantly. Only 16% of subjects with dementia with ADNC displayed pure pathology. In a setting with a low frequency of autopsies, the availability of a harmonized cohort of cases suitable for a certain study is thus poor. The lack of harmonization, however, may lead to significant difficulties in interpreting various study results. The observation of frequent concomitant NC is of major concern, particularly now that pharmacological treatment directed to Aβ has been initiated [43].

The percentage of subjects with clinically observed dementia increased from 2% in HPτ limited to LC, 7% in PART, and 28% in subjects with ADNC when all 1610 subjects were included in the analysis. Similarly, the incidence of concomitant altered protein pathologies increased from 19% in subjects with HPτ limited to LC to 41% in PART and 61% in ADNC. These observations indicate that subjects with ADNC do indeed frequently display concomitant alterations in their brain in addition to HPτ and Aβ, suggesting that MIXED-NC is more common than might have been anticipated. This is also seen in Table 2 and Table 3, as only 16% of all dementia subjects, when compared to 56% of subjects without dementia, display pure ADNC or PART. It is noteworthy that including only Braak stages I–IV and comparing ADNC (*n* = 793) with PART (*n* = 419), significantly (FE < 0.001) more subjects with ADNC, when compared to PART, displayed concomitant pathology (57 respective 41%). Moreover, significantly (FE < 0.003) more ADNC subjects (12%) displayed two concomitant pathologies (LATE-NC and LBD-NC) compared to PART (7%). This aligns with recent observations and supports the outcome here that in aged subjects and subjects with dementia with ADNC, concomitant alterations are common [17,18,44]. Whether this outcome is influenced by a selection bias, considering that only a handful of individuals with dementia arrive at autopsy, is impossible to comment on. In 51% of subjects with dementia with ADNC, one concomitant pathology was observed, and in 28%, two concomitant pathologies, were observed. The biological relationship or eventual dependency between these proteins is not clear. What will happen to other protein alterations when Aβ is efficiently deleted, i.e., HPτ, TDP43, and αS, all of which seem to correlate in extent with age and with each other? It is suggested that treatment needs to be initiated as early as possible, but even in the population without dementia, concomitant alterations were observed. Based on the obscured correlations and even negative correlation between the extent of HPτ, Aβ, αS, TDP43, and age (Table 6) in individuals with dementia, it would not be surprising if anti-Aβ treatment would, over time, lead to an increased number of individuals with dementia with PART in Braak stages V–VI, or even to a new type of dementia disorder.

### 3.7. Vascular Neuropathologic Change (VNC)

The incidence of VNC was 53% in HPτ limited to the LC, 53% in PART, and 69% in subjects with ADNC. Severe VNC was observed less often (HPτ limited to LC in <1%; PART in 1%; ADNC in 6%). Congruent with the above, VNC related to high blood pressure, such as etat lacunaire/etat crible, were observed in 18% of subjects with HPτ limited to the LC, 24% of subjects with PART, and 26% in ADNC. The increase in these alterations from HPτ limited to LC to ADNC is probably related to the age of the patient. VNC and severe VNC increased significantly (PCS < 0.001) with age. Interestingly, gross infarcts were observed in 24% of the subjects in the ninth decade but only in 18% of subjects in the seventh decade. A similar outcome was noted for microscopic infarcts, observed in 37% of subjects in the ninth decade compared to 20% observed in the seventh decade. Since 1960, subjects with high blood pressure have been treated with various medications, including β-blockers, and hypertension is one of the strongest modifiable cardiovascular risk factors [45,46]. In a country like Sweden with a governmental healthcare system, a large proportion of subjects with cardiovascular diseases have been treated. Consistent with the above, in 2002, diseases of the circulatory organs were assigned as the primary cause of death in close to 50% of the population, whereas in 2020, the percentage was only 28 (Swedish Health Ministry, Sweden Statistics). Treatment of cardiovascular diseases probably explains why subjects in the 10th decade (in 1960, they were in their 4th decade) displayed all types of VNC more often compared to subjects in the 7th decade (in 1960, they were in their 1st decade). There were only 19 subjects with dementia with ADNC and other protein alterations that also displayed severe VCN. It is noteworthy that none of our 293 subjects with dementia displayed pure VNC. It should, however, be kept in mind that even mild VNC might influence CD, for example, in cases with many small infarcts in the brain or with infarcts in a brain region of significance. A major obstacle regarding assessment of VNC is the lack of consensus criteria used by many. The recent assessment recommendations by Skrobot and colleagues have been cited only five times since publication in 2016 [19]. Furthermore, an agreeable definition on VNC seen in the brain in subjects with diabetes mellitus is missing [47]. Thus, the VNC in the aged population need to be further addressed by both caregivers and neuropathologists. In clinical settings and in some large population-based studies, hypertension (etat crible/etat lacunaire alterations) and cardiovascular disease (infarcts) are repeatedly suggested as risk factors for dementia, particularly for AD [48,49]. It needs to be kept in mind that these studies are based on clinical evaluation during life, and the definite diagnosis assessing all alterations in the brain postmortem that are of interest for CD, is mostly lacking.

### 3.8. Age-Related Tau Astrogliopathy (ARTAG)

ARTAG was described in 2015 as a glial HPτ alteration [19]. In our sample, we looked for this alteration in the basal forebrain and hippocampus sections and detected this NC in 37% of the 1610 subjects. ARTAG was observed in ADNC (FE < 0.001) significantly more frequently (43%) compared to PART (36%) and observed more often in subjects with dementia (53%) compared to subjects without dementia (38%). ARTAG increased significantly with the Braak stage of HPτ (Braak I 30%, Braak II 39%, Braak III 52%, Braak IV 57%, Braak V 47%, Braak VI 58%) and increased with age (7th decade 25%, 8th 37%, decade 9th decade 48%, 10th decade 69%, 11th decade 71%). In summary, ARTAG can be expected to be found in the brain of an aged dementia subject with ADNC. Interestingly, ARTAG was observed in 45% of subjects with VNC compared to 34% of subjects lacking VNC (FE < 0.001), and in 60% of subjects with severe VNC compared to 40% of subjects lacking severe VNC (FE = 0.005). Thus, ARTAG might be associated with VNC, and this association needs to be investigated further.

### 3.9. Definite Diagnoses

The most common definite neuropathological diagnosis in our aged postmortem dementia sample, including 293 subjects, was ADNC, assigned to 231 subjects (79%) (high level *n* = 102; intermediate level = 129). This observation is certainly in line with the clinical observation that AD syndrome is the most common one in the aged population [50]. In only eight (3%) subjects, the primary cause of dementia was PART (high level *n* = 1; intermediate level *n* = 7), i.e., cases with tangle-predominant dementia. It is noteworthy that only a handful of subjects with ADNC/PART (20%) displayed pure ADNC pathology. With age, the incidence of pure ADNC/PART decreased from 50% in the 6th decade to 12% in the 9–10th decade, and with the Braak stage, the incidence of pure ADNC decreased from 24% to 12%. The existence of and the relatively high frequency of MIXED-NC have been discussed previously. Already in 2007, prior to the observation of TDP43 pathology in the brain, Schneider and colleagues reported that MIXED brain pathologies are probably the most common cause of CD in older persons [51]. The concomitant alterations that were frequently observed in our study were LATE-NC, LBD-NC, and VNC. Similar outcomes, i.e., the common observation of concomitant NC with ADNC, have been reported by others [17,18]. In light of the above, it is surprising that a neuropathological assessment of aged subjects with CD, as seen here, is carried out relatively seldom. This is certainly alarming, as the clinical to neuropathological correlation is not 100% [52]. The agreement rates are certainly influenced by the stringency in the clinical assessments during life; not all subjects with CD are clinically assessed by neurologists, psychiatrists, or geriatricians when all available assessment techniques are implemented. Even if the clinical assessment is stringent, some of the alterations are more or less difficult to ascertain during life.

The second most common definite neuropathological diagnosis assigned in 34% of the subjects with dementia was LATE-NC, a relatively new entity [15]. Notably, LATE-NC was observed in few subjects without HPτ pathology and seldom in subjects with a low level of PART/ADNC (3% of all with ADNC/PART/LATE-NC). The correlation between the extent of HPτ and the extent of TDP43 was significant and strong. In 2020, while assessing 61 subjects with AD, McAleese and colleagues concluded that the presence of LATE-NC was not associated with an increase in HPτ or Aβ pathology [53]. Both the selection of cases, the number of cases in various age groups, and the methods used differ significantly; thus, the results are certainly not to be compared. The incidence of LATE-NC increased with the HPτ Braak stage from 37 to 46% in ADNC. Similarly, the incidence of LATE-NC increased with age, particularly in the PART cases, from 6% in the 6th decade to 60% in the 9th decade. Noteworthy, only 10% of subjects with ADNC and 2% of those with PART displayed severe LATE-NC, i.e., TDP43 pathology Josephs stage ≥3. Overall, there were only three subjects (1%) with severe TDP43 pathology Josephs stage ≥3 in combination with a low level of PART/ADNC (Braak stage I–II) and, in these subjects, dementia was presumed to be caused by LATE-NC. It is puzzling there were 14 subjects in our non-dementia cohort with a similar pathology, but as commented before, an eventual mild-to-moderate CD might not always have been assigned by the caregivers, or these subjects might have had a high individual resilience to CD [24]. Notably, a substantial number of subjects with dementia (27%) displayed both LATE-NC and LBD-NC. The function of TDP43 in the setting of neurodegeneration is still unclear. The protein was identified in 2001 as a human cystic fibrosis transmembrane conductance regulator exon 9 [54]. It is known to be expressed in the nuclei in somatic cells, not only in the brain. Since 2006, it has been known that this protein is the major culprit in FTLD; thus, when FTLD has been presumed clinically, the assessment of this alteration is a rule [55,56]. When assessing subjects with clinical diagnoses of AD or LBD, the assessment of TDP43 is not always carried out. What was observed here is that TDP43 pathology was common in the brain, and the extent was strongly associated with the extent of HPτ. The initiation site of TDP43 pathology, as seen in LATE-NC based on current knowledge, is the amygdala—a brain region that is, in principle, affected by all currently acknowledged protein alterations at a relatively early stage of their progression [12,13,14]. In light of this, that one alteration might influence the development of another, i.e., seeding, has been discussed [57].

The third most common NC observed in 21% of all cases was LBD-NC. Notably, as shown in Table 2 and Table 3, LBD-NC was never assigned in a case lacking HPτ pathology, and in one case with concomitant Ag. Severe αS pathology Braak stage ≥4 was observed in 16% of the whole cohort. The incidence of LBD-NC in the whole cohort increased with Braak stages (low 10% to high 41%). This was also seen in the strong positive correlation between the extent of HPτ and αS. Many of the subjects with dementia (21%) with severe LBD-NC (Braak stage ≥ 4) displayed concomitant ADNC rather than PART (12%); thus, a large number of subjects with clinical LBD syndrome would have had, during life (if performed), an Aβ-positive outcome by means of Pittsburgh Compound B (PiB) binding imaging [58]. This imaging outcome might mislead the healthcare provider, as clinicians primarily associate neocortical Aβ with AD rather than LBD syndromes [1]. If the PiB analysis would have been implemented prior to death as a diagnostic tool, only seven (21%) out of our thirty-three subjects with severe αS and concomitant PART or HPτ solely in the LC would have been diagnosed as suffering from dementia with Lewy bodies (DLB). Four subjects (12%) with concomitant low level of ADNC and thus an Aβ-positive outcome with PiB might have been clinically assigned as suffering from AD, in line with the subjects with concomitant severe ADNC and severe αS pathology, i.e., MIXED-NC (*n* = 22, 67%). The above certainly explains why the clinical and neuropathological correlations are poor, particularly in subjects with both dementia and movement disorder [52].

There were 192 subjects with MIXED NC, i.e., ADNC/PART with LATE-NC, LBD-NC, or both, constituting 64% of all 293 dementia cases. The assignment of subjects displaying MIXED-NC has varied significantly during the last few years [11,17,18,51]. Notably, based on our results, concomitant alterations were influenced by age; thus, in some centers, the number of subjects with MIXED pathologies might even be higher than reported here [51]. It has been recommended for many years that all alterations observed in the postmortem brain are assessed and reported, whereas the interpretation, i.e., the estimation of significance and assessment of causalities, varies from center to center [8]. The plethora of pathologies is and will be even more important when any of the alterations is eventually targeted, as is currently the case with the ongoing anti-Aβ trials.

### 3.10. Intervening with the Development and Progression of the Common Protein Alterations

In summary, acknowledging that NC are common in the ageing brains and that their extent increases significantly with age, and that most subjects with dementia in the aged population display a combination of various alterations, is of importance. There is currently no curative or controlling treatment that can alter the evolution of all pathological lesions observed in the ageing brain. The following question arises: what can and should a caregiver interfere with, whether it is HPτ, Aβ, TDP43, αS, or VNC? Intervening with the development of VNC has, since the 1970s, been efficient due to the development of pharmacological treatments for hypertension and cardiovascular diseases and should certainly be continued. Diabetes mellitus is a major contributor to vascular alterations and should certainly be treated accordingly [59,60]. Interventions consistent with anti-Aβ therapies have been ongoing for 20 years without major clinical benefits in large clinical trials [61]. The recently developed therapies using monoclonal Aβ-antibodies have shown promise with registered clinical efficacy, but the follow-up time is still somewhat short [62]. Intervening with HPτ might be difficult, as this alteration is seen at a relatively young age, and the detection of the pathology prior to symptoms, HPτ in the LC or entorhinal cortex, is currently difficult if not impossible [1,63]. Regarding αS, a recent publication reported that subjects who had undergone an appendectomy were less prone to develop clinical LBD [64]. This observation has been challenged and needs to be critically evaluated [65]. However, if valid, an anti-inflammatory treatment might be of interest. Notably, inflammation has been reported to be associated with many neurodegenerative diseases, including Alzheimer disease, Parkinson disease, amyotrophic lateral sclerosis, and multiple sclerosis [66]. Regarding TDP43, a positive association between TDP43 in the brain and amylin in the pancreas was reported to be found in the aged, with the latter, amylin, being a protein related to glucose metabolism that accumulates in the pancreas in diabetics [67]. This observation suggests than an association might be found between TDP43 and diabetes and should certainly be further explored in large cohorts.

## 4. Material and Methods

This study was conducted using postmortem brain tissue obtained during autopsies performed after receiving a referral from a caregiver to assess the cause of death and disease status. The autopsies were carried out at Uppsala University Hospital over a period of 15 years. The study was approved by the local ethical committee (Dnr 2011/286 and Dnr 2020/4184).

Each autopsy followed standard procedures, and the postmortem delay was noted. Samples from most parenchymal organs were collected, fixed in 4% buffered formalin, and paraffin-embedded for diagnostics and eventual future use. Based on the outcome of the postmortem examination, the pathologist assigned the cause of death. The brain was removed, weighed, and placed in 4% buffered formalin. After 2 to 5 days of fixation, gross examination was carried out by one of the two available neuropathologists in a standardized manner. The severity of arteriosclerosis was registered (none, mild, moderate/severe), and macroscopic lesions observed on coronal sections were noted. A standard set of brain tissue was sampled (neuroanatomical regions listed in Appendix A) and placed in commercial mega cassettes for further fixation in 4% buffered formalin. After an additional 1–2 weeks of fixation, the automatic paraffin embedding procedure was initiated.

Automated systems were used to stain the brain section, cut at 7 µm. The antibodies and methods applied for the immunohistochemical (IHC) stains to visualize altered proteins are listed in Appendix A. The antibodies and assessment strategies used have been shown to yield reproducible results in multi-center studies conducted by BrainNet Europe [68,69]. In the brain tissue, VNC was assessed via a routine histochemical stain, with hematoxylin-eosin-stained sections at magnifications of 20× and 100×.

Microscopic infarcts, regardless of the size or type, were identified in every hematoxylin-eosin-stained section, and etat crible/etat lacunaire alterations were assessed in the striatum. CAA was noted in the IHC-stained sections, applying an antibody directed to Aβ protein. ARTAG was registered as seen or not seen in the sections of basal forebrain and the section of the hippocampus, and argyrophilic grains (Ag) were noted in the hippocampal section while applying the antibody directed to HPτ protein (Appendix A) [70]. The staging of the visualized protein alterations in the IHC-stained section (Figure 3, Figure 4, Figure 5 and Figure 6) was determined based on the current consensus criteria, i.e., Braak stage for HPτ, Thal phase for Aβ, Josephs stage for TDP43, and Braak stage for αS [4,10,13]. In Table 7, the implemented staging strategies are summarized.

When relevant, to confirm a diagnosis of PSP/CBD, FTLD, or Ag disease, additional IHC stains applying antibodies directed to 3R and 4R tau were applied [71]. Subjects with gross and/or microscopic infarcts and/or etat crible/etat lacunaire alterations in the striatum were noted as displaying VNC. For a diagnosis of severe VNC considered sufficient to influence cognition, the subject had to have displayed gross or microscopic infarct in combination with etat crible/etat lacunaire lesions in the striatum, as well as cerebral amyloid angiopathy (CAA) in some of the assessed brain sections (parietal cortex, basal forebrain, midbrain, cerebellum).

For statistical analysis, IBM SPSS was used, applying non-parametric tests. For descriptive statistics, the mean ± standard error of the mean (m ± SE) was given. Statistical differences between the studied groups were assessed using the Mann–Whitney U (MWU) test. For the contingency of categorical data, Fisher’s exact (FE) test and Pearson chi-square (PCS) test were used. The correlation between the studied variables was assessed using the Spearman correlation test.

## 5. Conclusions

The ageing brain displays a plethora of NC, and when in excess, the NC are associated with CD. The incidence of NC increases with age, particularly from the eighth decade onward. The conversion of a normal subject to an individual with MCI might be caused by one or even two protein alterations, whereas the conversion of a subject with MCI to more severe CD dementia is primarily caused by mixed pathologies. The most common NC in the aged population are ADNC/PART, followed by LATE-NC and LBD-NC. Since MIXED-NC are the most common cause of CD in the elderly, intervening with only one protein alteration might lead to a disappointing outcome, i.e., the progression of other protein alterations or even the evolution of a new disease entity. A substantial number of subjects displayed HPτ limited to LC that might be stationary, progress to PART/ADNC, or progress to a primary tauopathy. The presence of VNC and severe VNC is common in subjects older than 80 at the time of death. However, their importance in relation to CD may decrease over time due to successful interventions targeting the causative mechanisms, i.e., hypertension and cardiovascular disease. The type of VNC to be looked for in diabetics needs to be elucidated. ARTAG is a common alteration in the brains of the elderly, but its significance is still unclear.

Overall, in order to address the issues related to numerous ageing-related brain alterations that may, when moderate or severe, cause CD, a postmortem assessment of brains needs to be facilitated. Furthermore, support for the sampling of brain tissue for use in research is crucial. Many questions are and will be answered through the implementation of cell cultures and animal models, by means of positron emission tomography, or through the assessment of biomarkers in cerebrospinal fluid and/or blood, but studies need to be carried out on human brain tissue when exploring ageing-related alterations influenced by various events, more- or less-severe systemic diseases, and even medications.

## Figures and Tables

**Figure 1 ijms-25-04065-f001:**
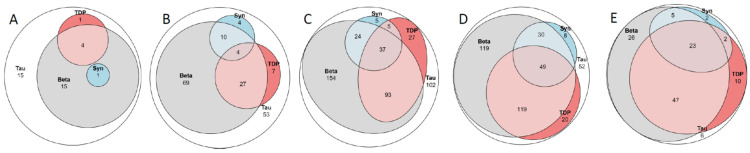
Venn diagram. Tau, hyperphospohorylated τ; Beta, amyloid β-protein; Syn, α-synuclein; TDP, transactive DNA-binding protein 43. (**A**) 36 subjects death in the sixth decade; (**B**) 179 subjects death in the seventh decade; (**C**) 447 subjects in the eight decade; (**D**) 400 subjects in the ninth decade and (**E**) 121 subjects death in the tenth to eleventh decade. See also Table 5.

**Figure 2 ijms-25-04065-f002:**
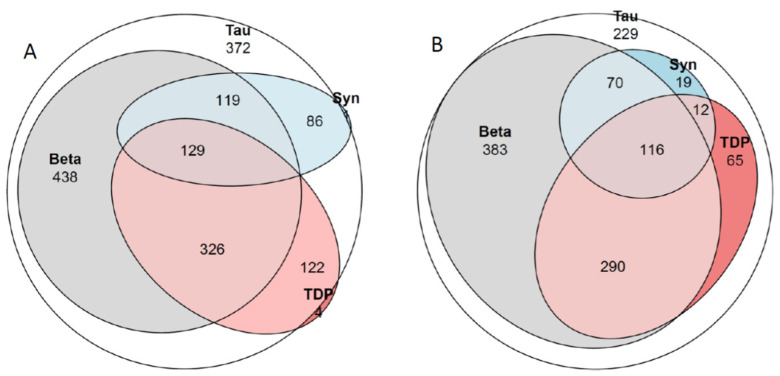
Venn diagram. Tau, hyperphospohorylated τ; Beta, amyloid β-protein; Syn, α-synuclein; TDP, transactive DNA binding protein 43. (**A**) all 1610 subjects. There were 372 subjects with only HPτ, 3 subjects with only Aβ, 1 subject with only αS and 4 subjects with only TDP43. In (**B**) 1184 subjects with Alzheimer disease neuropathologic change (ADNC) and Primary age related tauopathy (PART). There were 229 subjects with only HPτ pathology. See also Table 2, Table 3 and Table 4.

**Figure 3 ijms-25-04065-f003:**
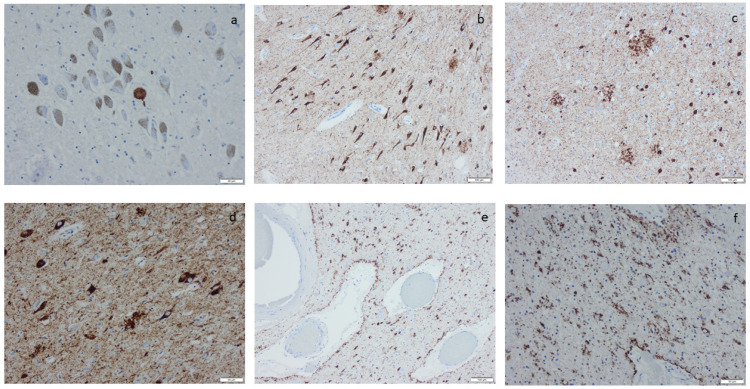
Hyperphosphorylated τ (HPτ) visualized by means of immunohistochemistry, applying antibodies as listed in Appendix A: (**a**) a solitary neuron with HPτ pathology surrounded by unaffected pigmented neurons; (**b**) numerous labeled “tangles” in the cornu ammonis region I of the hippocampus; (**c**,**d**) tangles, neurites, and neuritic plaques in the amygdala; (**e**,**f**) labeled glial cells as seen in age-related tau astrogliopathy. Scale bar: 100 µm in (**b**,**c**,**e**) and 50 µm in (**a**,**d**,**f**).

**Figure 4 ijms-25-04065-f004:**
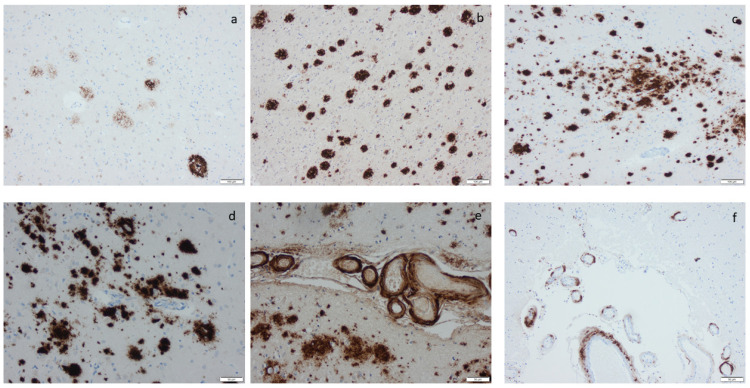
Amyloid β-protein (Aβ) visualized by means of immunohistochemistry, applying antibodies as listed in Appendix A: (**a**) diffuse aggregates in amygdala, (**b**–**d**) compact aggregates and plaques of various size in the cortex, (**e**) cerebral amyloid angiopathy (CAA) in the meninges in a case with concomitant Aβ aggregates in the tissue, and (**f**) CAA in the meninges in a case lacking Aβ in the tissue. Scale bar: 100 µm in (**a**,**b**,**c**) and 50 µm in (**d**,**e**,**f**).

**Figure 5 ijms-25-04065-f005:**
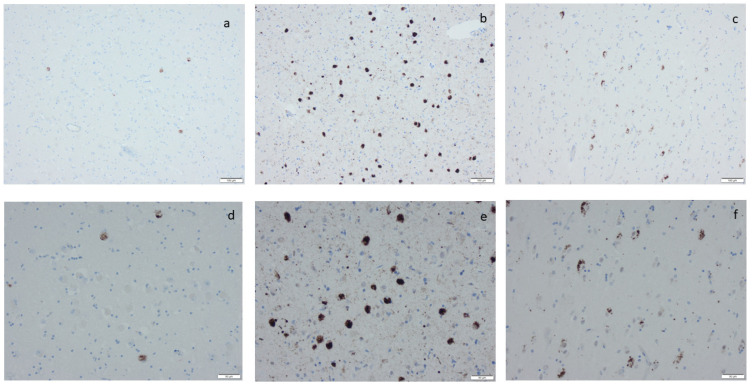
Phosphorylated transactive DNA-binding protein 43 labeling by means of immunohistochemistry, applying antibodies as listed in Appendix A: (**a**,**d**) labeled inclusions in the amygdala, (**b**,**e**) numerous labeled inclusions in the amygdala, and (**c**,**f**) labeled inclusions in the hippocampus as seen in limbic-predominant age-related TDP encephalopathy. Scale bar: 100 µm in (**a**–**c**) and 50 µm in (**d**–**f**).

**Figure 6 ijms-25-04065-f006:**
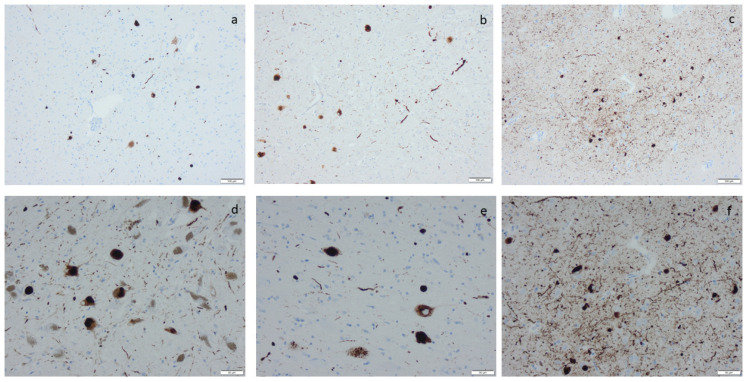
α-synuclein labeling by means of immunohistochemistry, applying antibodies as listed in Appendix A: (**a**) labeled inclusions, i.e., Lewy bodies in motor nucleus of vagus; (**b**) Lewy bodies and neurites in nucleus basalis of Meynert (**c**,**f**), in the amygdala (**d**), in the substatia nigra, and (**e**) in the locus coeruleus. Scale bar: 100 µm in (**a**–**c**) and 50 µm in (**d**–**f**).

**Table 1 ijms-25-04065-t001:** Flowchart. ADNC, Alzheimer disease neuropathologic change; PART, primary age-related tauopathy; TDP43, transactive DNA-binding protein 43.

36,034 deceased subjects in the morgue during 15 years		
4389 (12% of the deceased) autopsies performed		
1630 (37% of the autopsied and 5% of the diseased) neuropathological assessments carried out		
1610 of the subjects age at death ≥41 years		
1576 subjects displayed hyperphosphorylated τ in the locus coeruleus or with the distribution as seen in ADNC/PART		34 subjects 14 lacked neuropathological changes 3 subjects displayed only amyloid β-protein in the brain 17 displayed primary tauopathies
1421 subjects displayed hyperphosphorylated τ in the locus coeruleus or with the distribution as seen in ADNC/PART		155 subjects displayed hyperphosphorylated τ in locus coeruleus or in Braak stage I–II and concomitant significant other neuropathologic change 113 subjects with α-synuclein Braak stage ≥ 4 [10] 26 subjects with TDP43 pathology Josephs stage ≥ 3 [13] 21 subjects with frontotemporal lobar degeneration
1184 subjects with hyperphosphorylated τ in Braak stages I–VI	237 subjects with hyperphosphorylated τ limited to the locus coeruleus	

**Table 2 ijms-25-04065-t002:** Neuropathologic changes (NC) in 1317 subjects without dementia aged ≥41 years old at the time of death, with mean age ± standard error of means of 74.8 ± 0.3, comprising 56% males. HPτ, hyperphosphorylated τ; PART, primary age-related tauopathy; ADNC, Alzheimer disease neuropathologic change; L, low; I, intermediate; H, high level of PART or ADNC [6,8]; ARTAG, age-related tau astrogliopathy; CAA, cerebral amyloid angiopathy; FTLD, frontotemporal lobar degeneration; LATE, limbic-predominant age-related TDP encephalopathy, transactive DNA-binding protein 43 Josephs stage ≤ 2 (a), ≥3 (b) [13]; LBD, Lewy body disease, α-synuclein Braak stage ≤ 3 (a), ≥4 (b) [10]; MSA, multi-system atrophy; PSP/CBD, progressive supranuclear palsy/corticobasal degeneration; VNC, vascular NC.

	No HPτ	with HPτ/LC	with PART		with ADNC				
Concomitant NC			L	I	L	I	H	∑	% of ∑
All cases	29	247	340	51	426	215	9	1317	
No concomitant NC	14	197	211	15	250	54	1	742	56
+LATE-NC a/b		22/1	45/3	14/2	74/11	80/17	3/2	274	21
+LATE-NC + CAA			7	2				9	<1
+LBD-NC a/b		10/7	14/22	2/1	21/34	6/18	1/0	136	10
+LBD + CAA			0/3	1				4	<1
+MSA + CAA			1					1	<1
+LATE-NC + LBD-NC		3	14	6	27	35	2	87	7
+Argyrophilic grains (Ag)	2	1		2	1	1		7	<1
+Ag + LATE-NC	1			2				3	<1
+Ag + LBD-NC a/b	1/0				1			2	<1
+Ag + LATE-NC + LBD-NC				1	1			2	<1
+Ag + PSP	1				1			2	<1
+PSP/CBD	5			1				6	<1
+PSP/CBD + LATE-NC	1				3	1		5	<1
+CBD + AgD + LATE-NC	1							1	<1
+FTLD			3		2	3		8	<1
+ARTAG	7	33	108	29	139	111	7	435	33
+VNC	11	130	181	25	280	149	7	783	59
+severe VNC		1	4	2	24	14		45	3

**Table 3 ijms-25-04065-t003:** Neuropathologic changes (NC) in 293 subjects with dementia aged ≥41 years old at the time of death, with mean age ± standard error of means of 80.3 ± 0.5, comprising 54% females. HPτ, hyperphosphorylated τ; PART, primary age-related tauopathy; ADNC, Alzheimer disease neuropathologic change; L, low; I, intermediate; H, high level of PART or ADNC [6,8]; ARTAG, age-related tau astrogliopathy; CAA, cerebral amyloid angiopathy; FTLD, frontotemporal lobar degeneration; LATE, limbic-predominant age-related TDP encephalopathy, transactive DNA-binding protein Josephs stage ≤2 (a), ≥3 (b) [13]; LBD, Lewy body disease, α-synuclein Braak stage ≤3 (a), ≥4 (b) [10]; MSA, multi-system atrophy; PSP/CBD, progressive supranuclear palsy/corticobasal degeneration; VNC, vascular NC.

	No HPτ	with HPτ/LC	with PART			with ADNC				
Concomitant NC			L	I	H	L	I	H	∑	% of ∑
All cases	5	4	20	9	1	17	135	102	293	
No concomitant NC							34	12	46	16
+LATE-NC a/b			0/2	1/0		0/1	36/14	31/16	101	34
+LATE-NC + CAA a/b				1/0	1/0				2	<1
+LBD-NC a/b		0/1	0/3	0/3		0/4	2/11	1/8	33	11
+LBD-NC + CAA a/b			0/1						1	<1
+MSA							1		1	<1
+LATE-NC + LBD-NC		1	6	2		5	31	34	79	27
+Argyrophilic grains (Ag)				1					1	<1
+Ag +LATE-NC+ CAA	1			1					2	<1
+PSP/CBD	4								4	1
+PSP + LATE-NC							1		1	<1
+PSP + LBD-NC						0/1			1	<1
+FTLD-		2	8			6	5		21	7
+ARTAG	2	1	14	6		8	76	48	155	52
+VNC		3	14	7		10	91	82	207	71
+severe VNC		1					12	4	17	6

**Table 4 ijms-25-04065-t004:** Demographics of 1184 subjects with primary age-related tauopathy (PART)-respective Alzheimer disease neuropathologic change (ADNC). ARTAG, ageing-related tau astrogliopathy; HPτ, hyperphosphorylated τ; LBD, Lewy body disease; LATE, limbic-predominant age-related TDP encephalopathy; VNC, vascular neuropathologic change; MWU, Mann–Whitney U test; FE, Fisher’s exact test.

	ALL *n* (%)	PART *n* (%)	ADNC *n* (%)	Statistics
All	1184	362 (31)	822 (69)	
Male/female	652/532	212/150	440/382	Ns
Age range	48–102	48–102	50–102	
Mean age at death ± standard error	77.8 ± 0.3	75.8 ± 0.5	78.7 ± 0.3	MWU < 0.001
Eith dementia	248 (21)	11 (3)	237 (29)	FE < 0.001
Brain weight in grams	1351 ± 5	1373 ± 9	1341 ± 6	MWU < 0.002
With HPτ, Braak stages I–II [2]	661 (56)	301 (83)	361 (44)	FE < 0.001
With HPτ, Braak stages III–IV [2]	410 (35)	60 (17)	350 (43)	FE < 0.001
With HPτ, Braak stages V–VI [2]	112 (10)	1 (<1)	111 (14)	FE < 0.001
With ARTAG	487 (41)	130 (36)	357 (43)	FE 0.018
With argyrophilic grains	10 (1)	7 (2)	3 (<1)	FE 0.012
With LATE-NC (Josephs ≥1) [13]	482 (41)	93 (26)	389 (47)	FE < 0.001
With severe LATE-NC (Josephs ≥3) [13]	83 (7)	2 (<1)	81 (10)	FE < 0.001
With LBD-NC [10]	216 (18)	36 (10)	180 (22)	FE < 0.001
With severe LBD-NC (Braak stage ≥4) [10]	136 (12)	11 (3)	125 (15)	FE < 0.001
With cerebral amyloid angiopathy (CAA)	388 (33)	37 (10)	351 (43)	FE < 0.001
Hippocampal sclerosis (HS)	32 (3)	8 (2)	24 (3)	
With V-NC	759 (64)	191 (53)	568 (69)	FE < 0.001
With severe V-NC	55 (5)	5 (1)	50 (6)	FE < 0.001

**Table 5 ijms-25-04065-t005:** Percentage of cases with alterations in subjects with primary age-related tauopathy (PART)-respective Alzheimer disease neuropathologic change (ADNC) in five age groups. ARTAG, ageing-related tau astrogliopathy; CAA, cerebral amyloid angiopathy; LATE-NC, limbic-predominant age-related TDP encephalopathy—neuropathologic change; LBD, Lewy body disease, VNC, vascular NC.

Age at Death in Years	50–59		60–69		70–79		80–89		90–102	
Lesions	PART	ADNC	PART	ADNC	PART	ADNC	PART	ADNC	PART	ADNC
	36		179		447		400		121	
Number	16	20	71	108	155	292	94	306	25	96
Male/female	10/6	11/9	48/23	64/44	90/65	152/150	54/40	160/146	9/16	53/43
Results given in percent and difference when comparing ADNC with PART cases ^Fischer’s exact test^										
With dementia		10		8^0.012^	5	24^<0.001^	1	41^<0.001^	12	32^0.049^
+HPτ, Braak stages I–II [2]	100	75^0.053^	92	75^0.006^	87	54^<0.001^	76	28^<0.001^	56	25^0.007^
+HPτ, Braak stages III–IV [2]		15	9	18	13	33^<0.001^	25	58^<0.001^	44	56
+HPτ, Braak stages V–VI [2]		10		7^0.023^	<1	13^<0.001^		15^<0.001^		19^0.023^
+ARTAG	6	10	25	25	34	39	46	48	64	71
+Argyrophilic grain			1		1	0	3	<1	4	1
+LATE-NC (Josephs ≥1) [13]	6	20	16	30^0.033^	25	42^<0.001^	29	54^<0.001^	60	69
+LATE-NC (Josephs ≥3) [13]				3	<1	7^0.002^	1	14^<0.001^		17^0.041^
+LBD-NC (Braak stage ≥1) [10]		5	9	16	8	20^<0.001^	14	25^0.024^	20	27
+LBD-NC (Braak stage ≥4) [10]			1	6	3	13^<0.001^	4	20^<0.001^	8	22
+CAA		25^0.053^	6	32^<0.001^	10	40^<0.001^	13	48^<0.001^	20	51^0.006^
+V-NC	44	70	42	52	51	65^0.004^	61	76^0.006^	75	79
+Severe V-NC		5		2	1	5	3	8		9

**Table 6 ijms-25-04065-t006:** Spearman’s rho (r) correlation between different neuropathologic changes and ages. The 1184 subjects described in Table 4 included significance ^1^ 0.05, ^2^ 0.01 and ^ns^ not significant. PART, primary age-related tauopathy; ADNC, Alzheimer disease neuropathologic change; TDP43, phosphorylated transactive DNA-binding protein 43—extent TDP43 given as Josephs stage [13], HPτ extent given as Braak stage [2], Aβ extent given as Thal phase [4], and αS extent given ar Braak stage [10].

	All	PART	ADNC	Dementia	Non-Dementia
Number	1184	362	822	248	936
Hyperphosphorylated (HP) τ with age	0.36 ^2^	0.28 ^2^	0.33 ^2^	−0.16 ^1^	0.35 ^2^
Amyloid β-protein (Aβ) with age	0.22 ^2^		0.22 ^2^	−0.06 ^ns^	0.16 ^2^
TDP43 with age	0.26 ^2^	0.21 ^2^	0.25 ^2^	0.14 ^ns^	0.25 ^2^
α Synuclein (αS) with age	0.14 ^2^	0.15 ^2^	0.12 ^2^	0.13 ^ns^	0.11 ^2^
Aβ with HPτ	0.62 ^2^		0.66 ^2^	0.59 ^2^	0.43 ^2^
TDP43 with HPτ	0.50 ^2^	0.29 ^2^	0.51 ^2^	0.32 ^2^	0.41 ^2^
αS with HPτ	0.33 ^2^	0.23 ^2^	0.32 ^2^	0.06 ^ns^	0.26 ^2^
TDP43 with Aβ	0.39 ^2^		0.43 ^2^	0.32 ^2^	0.26 ^2^
αS with Aβ	0.28 ^2^		0.29 ^2^	0.11 ^ns^	0.18 ^2^
TDP43 with αS	0.21 ^2^	0.12 ^1^	0.20 ^2^	0.10 ^ns^	0.14 ^2^

**Table 7 ijms-25-04065-t007:** Staging criteria implemented while assessing the extent of altered proteins visualized by immunohistochemistry.

Altered Protein	Staging Criteria	Stages	Based on the Visualization of the Altered Protein in
Hyperphosphorylated τ	Braak stage [2]	I to VI	The hippocampus reaching towards occipital cortex
Amyloid β protein	Thal phase [4]	1 to 5	The neocortex reaching towards cerebellum
α synuclein	Braak stage [10]	1 to 6	The medulla, nucleus vagus, reaching towards the parietal/frontal cortices
Transactive DNA-binding protein 43	Josephs phase [13]	1 to 5	The amygdala reaching towards frontal cortex

## Data Availability

Data is not available due to privacy and ethical restrictions.

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
