# Peer review of "Ageing-Related Neurodegeneration and Cognitive Decline"

_ijms, 2024, doi:10.3390/ijms25074065_

Round 1
Reviewer 1 Report
Comments and Suggestions for Authors
The manuscript by Irina Alafuzoff and Sylwia Libard is a detailed and well written text covering the pathological changes seen in postmortem brain of large cohort studied over 15 years. I have a few suggestions that I hope will help authors improve their manuscript.
1. Despite being the commonly used and commonly accepted “misnomers”, the eponymous terms “Parkinson’s Disease” and “Alzheimer’s Disease” (AD) are logically incorrect. Historically, the diseases were discovered by Charles Parkinson or Alois Alzheimer, respectively; the diseases were not “their own” diseases. Because of the eponymous convention, using the possessive form (apostrophe plus “s” or genitive “s”) is wrong but has been perpetuated in the English Scientific literature by our great peers. Many though have avoided it [1-2]. The Australian Manual of Scientific Style and The Chicago Manual of Style also advise against the use of the possessive form. I suggest taking their editorial advice and applying it throughout the text.
2. Please change the terminology used to describe Aβ before using this abbreviation. The WHO-designated terminology for this protein is “amyloid β-protein”. Please use this terminology throughout.
3. Please revise the naming of the two US organisations, “National Institute on Aging” and “Alzheimer's Association”. Specify that these organisations belong to US, not “National” to every other country.
4. Include a summary table of all the different staging criteria (e.g. Braak, TDP-43, etc.) will be informative to the readers, avoiding reference to the original manuscripts. Please add this information.
5. Remove the abbreviations that are unnecessary and not repeated. Conventionally, a term must be used 4–5 times before its abbreviation can be introduced. Generally, please reduce the numbers of the abbreviations used in the text to ease understanding of the text by the audience. Some abbreviations like “CD”, “NC”, and “PM” are unnecessary because they’re either two words or one. To ease reading and understanding, reduce the profuse use of the abbreviations throughout the paper.
6. In lines 28, 107, 140, 369, 371: remove the unreadable, cryptic, characters after “HP”.
7. Some abbreviations are not defined; see line 18 for αS. Some are hyphenated unnecessarily; see line 25 for V-NC. Use VNC instead and revise similar constructions throughout the text.
8. Line 86: How was the cause of death “determined”? Do you mean “documented” by referring the medical history of the cases instead of “determined”?
9. Line 87: Change “grossing” to “gross examination”.
10. The authors seem to tend to use excess nominalisations (use of a noun form of a verb instead of the verb). Please remove these and rephrase. For example, in line 94: “All stains were performed …” can be changed to “Automated systems were used to stain the brain sections, cut at 7 µm.”
11. Define IHC.
12. Line 99: revise the use of the letter x and capital M.
13. Add the missing punctuation mark in line 126; revise throughout the text.
14. Revise the statement in lines 137–138: “Following …”
15. Complete the sentence in line 158 and line 204. “ … excluded from the following …” What? A concept is missing after “following”.
16. Number all the tables in the text so cross-referring to them is clear.
17. What does “a/b” mean in Table on page 4 and Table on page 5? Clarify.
18. In line 178, place the parenthetic descript after “subjects”.
19. In lines 186 and 187, correct the run-on sentence with the comma splice or comma fault.
20. In 191, write FTLDNC or FTLD-NC, not FTLD- with the hanging hyphen.
21. Revise all the constructions like the one used in line 241. The subject was not 59 years old. The subject’s death occurred at that age. Please be precise and informative.
22. In line 222, one capitalisation is unnecessary. Please remove.
23. Line 246: remove hyphens after the numbers inside the parenthesis. And generally, throughout the text, revise the conventional use of the numbers: numbers 1–9 are conventionally spelled out in academic writing unless they’re followed by and SI unit of measurement of the percentage sign (for example). Revise for the entire text.
24. Revise the numbering of the subheadings in the text. These should be in sequence assumingly.
25. Line 254: does “CI” mean “MCI”? Revise.
26. Please produce and present Venn diagrams where understanding of the text can be reinforced by schematics; for example, when describing the cases’ overlapping pathologies in different age groups.
27. Revise use of “Noteworthy” in line 218 and other similar uses.
28. In line 362, do you mean 1,184, instead of 1,187?
29. In table on page 10, please remove the use of slash to indicate correlation or clarify this. Slash normally means division, separation, the word “or”, or a fraction. So, the usage of this punctuation mark in your paper is unclear or confusing.
30. Line 471: change “25,” to “25,000.” if this is what you mean.
31. It seems the use of “cohort” (line 471) may not be correct for this study because postmortem brain tissue was examined. Please revise.
32. Line 500: use “may have been” or “might have been”.
33. Revise line 522, 560, 619, and present a justification for it. Why do the authors assume that anti-Aβ treatment is needed while this treatment and the concept of amyloid hypothesis have been debated for years and clinical trials and medication based on anti-Aβ treatments keeps failing.
34. Related to this comment, please provide information on any type of treatment the studied deceased cases may have had during their lifetime or before dying.
35. Related to comment 33, please provide evidence of immunostaining for inflammatory markers in the brain. Why did the authors ignore the use of the inflammatory markers in the brain tissue? If this evidence is present, please discuss in the context of this research. If not, please discuss the limitations of this study.
36. Line 734: provide the missing information about the Joseph’s staging.
37. Line 705: do you mean TDP43 instead of TdP?
38. Line 711: why a different font and underlining are used?
References:
1. Arispe N, Pollard HB, Rojas E. Giant multilevel cation channels formed by Alzheimer disease amyloid β-protein [AβP-(1–40)] in bilayer membranes. Proc Natl Acad Sci USA. 1993;90(22):10573-7.
2. Arispe N, Pollard HB, Rojas E. β-Amyloid Ca(2+)-channel hypothesis for neuronal death in Alzheimer disease. Mol Cell Biochem. 1994;140(2):119-25.
Comments on the Quality of English LanguagePlease see my detailed comments. Reduce the use of abbreviations, avoid verb nominalisations, revise punctuation conventions, use complete sentences, revise conventions using numbers.
Reviewer 2 Report
Comments and Suggestions for Authors
The paper entitled "Ageing related neurodegeneration and cognitive decline", is interesting, very complete, and well structured. Compliments to the authors.
I have just a few suggestions:
1. Lines 135-168, including di table 1, could be moved, in the section Material and Methods. This part of the manuscript is more appropriate to collocate in the material than in the results.
2. The immunohistochemical study is morewide, an element that would enhance the paper would be to insert more histological photos.
It would be optimal to insert a photo for each antibody, composed of several images.
3. In supplementary material could be insert the negative control samples of each antibody.
Comments on the Quality of English LanguageMinor editing of English language required
